# Spatial, Temporal, and Geographical Factors Associated with Stranded Marine Endangered Species in Thailand during 2006–2015

**DOI:** 10.3390/biology12030448

**Published:** 2023-03-15

**Authors:** Pangram Pradip Na Thalang, Sukanya Thongratsakul, Chaithep Poolkhet

**Affiliations:** Section of Epidemiology, Faculty of Veterinary Medicine, Kasetsart University, Kamphaeng Saen, Nakhon Pathom 73140, Thailand

**Keywords:** Geographic Information System (GIS), marine endangered species, marine stranded, surveillance, Thailand

## Abstract

**Simple Summary:**

Understanding the stranding of endangered marine species is vital for their conservation. Against this background, we analysed stranding data from 2006–2015 in Thailand. A total of 1988 stranding events were obtained with an average of 198.80 stranded marine endangered animals per year. Of the recorded strandings, the majority were sea turtles, followed by dolphins, whales, and dugongs. Most strandings occurred along the coast of the Gulf of Thailand. The biggest clustering and highest density of stranded animals was observed in Phuket province. In general, the average number of animals stranded in the rainy season was significantly higher than that in the summer and winter. Our results indicate the importance of developing surveillance and monitoring systems to focus on and assist stranded animals as quickly as possible. This will enable and support related conservation efforts.

**Abstract:**

The stranding of endangered marine animals is cause for concern. We used spatial and temporal analyses to investigate the stranding of endangered marine species (whales, dolphins, dugongs, and sea turtles) in Thailand, based on stranding data and geographical records during 2006–2015. A total of 1988 stranding events were obtained, including 105 whales (5.28%), 714 dolphins (35.92%), 103 dugongs (5.18%), and 1065 sea turtles (53.57%), at an average of 198.80 stranded animals/year (standard deviation = 47.19). Most strandings occurred along the Gulf of Thailand (56.94%), while the rest occurred along the Andaman Sea (43.06%). Cluster and kernel analyses showed that strandings were the most common in Phuket Province. The average number of stranded animals in the rainy season was significantly higher than that in summer and winter (*p* < 0.05). Our results indicate that the coastline of Thailand was significantly associated with the number of seasonal strandings (*p* < 0.05). However, there was no significant association between environmental factors and the number of strandings. In conclusion, surveillance systems based on spatial and temporal analyses should be established to monitor stranded animals. This will help relevant authorities to rescue stranded animals more effectively and to study the causes of stranding.

## 1. Introduction

In Thailand, whales, dolphins, dugongs, and sea turtles are marine endangered species at risk of extinction. Most of these animals are classified as protected species according to the Wildlife Conservation and Protection Act, B.E. 2562 of Thailand and are included in the International Union for Conservation of Nature’s (IUCN) Red List of Threatened Species [1]. These marine species play an important role in the food chain and biodiversity and are essential for maintaining balanced and thriving ocean ecosystems [2,3]. Therefore, analysis of the data of these stranded animals can guide the establishment of a surveillance system for stranded marine endangered species. As a result, it will be possible to create control measures for reducing the loss of these animals, which affects the balance of marine ecosystems.

Various species of whales and dolphins are abundant in Thailand’s seas, namely, the Andaman Sea and the Gulf of Thailand. Examples of whale and dolphin species include the melon-headed whale (*Peponocephala electra*), ginkgo-toothed beaked whale (*Mesoplodon ginkgodens*), Irrawaddy dolphin (*Orcaella brevirostris*), Indo-Pacific humpback dolphin (*Sousa chinensis*), finless porpoise (*Neophocaena phocaenoides*), sperm whale (*Physeter macrocephalus*), and striped dolphin (*Stenella coeruleoalba*) [4,5,6]. According to a report by the Department of Marine and Coastal Resources (DMCR), approximately 500 whales and dolphins were present in Thailand’s seas in 2009 [7], while dugongs (*Dugong dugon*) were found in both the Andaman Sea and the Gulf of Thailand [8]. The DMCR had projected that the number of dugongs in the sea of Thailand would be approximately 240 in 2009 [9]. Sea turtles are found regularly in both the Gulf of Thailand and the Andaman Sea, particularly along beaches and islands. Examples of sea turtle species include leatherback sea turtles (*Dermochelys coriacea*), green turtles (*Chelonia mydas*), olive-ridley sea turtles (*Lepidochelys olivacea*), and hawksbill sea turtles (*Eretmochelys imbricata*). According to the DMCR, there were approximately 300 sea turtle nests on Thailand’s coastline in 2009 [10].

In Thailand, the primary causes of stranding of endangered marine species include topography, marine pollution, climate, natural toxicity, tidal disturbances, predator escape, chase victims, natural infection, misguided swarms, anthropogenic injuries, loss of echolocation, and navigation signalling problems [11]. Recently, fishing nets were reported to be associated with the stranding of sea turtles in the Gulf of Thailand. Researchers have also found macroplastic debris in the gastrointestinal tracts of stranded sea turtles [12]. In other countries, marine mammals have become stranded owing to illness, fisheries activities, and environmental changes [13], while sea turtle strandings have been reported to be related to anthropogenic factors and illness [14].

Marine strandings also include decomposed carcasses, making it impossible to determine the cause of death and stranding. This is also an obstacle to monitoring stranding and abnormalities in marine ecosystems. Notably, live stranded animals should be rescued immediately, but this requires the relevant authorities to have sufficient information on potential strandings at particular places and times. The application of geographic information systems to analyse the location and time of stranding can increase the chances of successful surveillance and monitoring. Therefore, this study aimed to use spatial and temporal analyses to understand the stranding of endangered marine species (whales, dolphins, dugongs, and sea turtles) in Thailand. In addition, geographical factors associated with stranding were investigated.

## 2. Materials and Methods

### 2.1. Definition of Marine Stranding, Data Collection, and Ethical Statement

In this study, we defined the stranding of endangered marine species as the appearance of whales, dolphins, dugongs, and sea turtles on the beach or on islands in the sea of Thailand. Both living and non-living animals were included in the study. For the study location, the coastal areas of Thailand were divided into two parts: the coastal area adjacent to the Andaman Sea and the coastal area adjacent to the Gulf of Thailand (Figure 1a). Thailand has 24 provinces connected to the sea, comprising six provinces connected to the Andaman Sea and 18 provinces connected to the Gulf of Thailand.

This study included the following data from 2006 to 2015: (1) the stranded data obtained from the DMCR, Thailand. This dataset includes animal species, dates, and coordinates of stranded animals, and animal/carcass characteristics. A total of 2261 animal stranding events were recorded; (2) Geographical factors that possibly affected the stranding of endangered marine species based on the Marine Knowledge Hub website [15]. These factors include the length of the coastline (km) and the number of islands in each province. In addition, in Thailand, the fisheries season is year-round, with fishermen always using similar equipment for different marine species. Therefore, data on the fisheries season are not available in this study.

This study used secondary data, which cannot be traced back to individual animals or people. Therefore, ethical approval was not required by the institutional review board. Moreover, this study followed the International Guiding Principles for Biomedical Research Involving Animals (1986).

### 2.2. Data Manipulation and Analysis

#### 2.2.1. Data Management and Descriptive Statistics

After data cleaning, 1988 of the 2261 obtained records were used for analysis. The remaining 273 records were incomplete because the stranding locations were not compatible with the given address or the stranding coordinates were not within the boundaries of Thailand. Data management and graphs were manipulated and displayed using Microsoft Excel 2016 (Microsoft Corp., Redmond, WA, USA). For temporal analyses, stranding points were divided into 10 subsets by year. In each year, the subset of stranding events was divided into three categories by season as follows: the rainy season (June to October), winter (November to February), and summer (March to May). Finally, 30 seasons were selected for further analyses.

#### 2.2.2. Spatial and Temporal Analyses

Stranding events during the 2006–2015 study period were mapped using ArcGIS release 10.3.1, and 10.8.2 (ESRI, Redlands, CA, USA). Kernel density and cluster analyses using Anselin Local Moran’s I were used to identify the density of stranding locations and clusters (clusters of high values or hot spots) [16] using ArcGIS 10.4.1 and 10.8.2. Thirty seasons of stranding locations for marine endangered species were also evaluated for clustering. Notably, all cluster analyses were based on an optimal grid size of 5 × 5 map units [17]. The average number of stranding locations was tested for statistical differences between the rainy, winter, and summer seasons using one-way analysis of variance (ANOVA) and Bonferroni tests by NCSS 2019 (NCSS, Kaysville, UT, USA). In addition, Pearson’s chi-square tests were used to evaluate the association between coastal location and season using NCSS 2019.

#### 2.2.3. Analysis of Association of Stranding Location and Geographical Factors

The correlation between the number of stranding locations and geographical factors, including the length of coastline (km) and number of islands, was measured using Spearman’s rank correlation (*Rs*) by NCSS 2019 at the provincial level. However, of the 1988 animal stranding locations, three events could not be linked to a particular province because they were found in the sea and could not be classified within the boundaries of any province. Therefore, the number of stranding events for this analysis was reduced to 1985.

## 3. Results

### 3.1. General Information

From the 1988 stranding events for marine endangered species, we found 105 whales (105/1988; 5.28%), 714 dolphins (714/1988; 35.92%), 103 dugongs (103/1988; 5.18%), and 1065 sea turtles (1065/1988; 53.57%) (Figure 1a). Notably, one event (1/1988; 0.05%) of a stranded animal was an unidentified species due to carcass decay. For whales, at least 13 categories were identified by common name as follows: Bryde’s whales (28/105; 26.67%), false killer whales (21/105; 20%), Omura’s whales (9/105; 8.57%), sperm whales (5/105; 4.76%), dwarf sperm whales (5/105; 4.76%), short-finned pilot whales (2/105; 1.90%), pygmy killer whales (2/105; 1.90%), pygmy sperm whales (2/105; 1.90%), Cuvier’s beaked whale (1/105; 0.95%), Blainville’s beaked Whale (1/105; 0.95%), killer whale (1/105; 0.95%), fin whale (1/105; 0.95%), blue whale (1/105; 0.95%), and unidentified (26/105; 24.76%). Notably, 20 whales (20/105; 19.05%) were alive. For dolphins, there were 11 categories by common name: finless porpoises (207/714; 28.99%), Irrawaddy dolphins (201/714; 28.15%), Indo-Pacific humpbacked dolphins (106/714; 14.85%), bottlenose dolphins (71/714; 9.94%), striped dolphins (42/714; 5.88%), spinner dolphins (34/714; 4.76%), spotted dolphins (22/714; 3.08%), rough-toothed dolphins (15/714; 2.10%), Risso’s dolphins (7/714; 0.98%), Fraser’s dolphins (6/714; 0.84%), and long-beaked common dolphins (3/714; 0.42%). Of these, 87 dolphins (87/714; 12.18%) were alive. For dugongs, 12 (12/103; 11.65%) were alive. For sea turtles, we found green turtles (543/1065; 50.99%), hawksbill sea turtles (277/1065; 26.01%), olive-ridley sea turtles (210/1065; 19.72%), leatherback turtles (11/714; 1.03%), loggerhead turtles (7/1065; 0.66%), and unidentified (17/1065; 1.60%). A total of 566 (566/1065; 53.15%) sea turtles were alive.

The average number of stranded marine endangered species was 198.80 animals/year (standard deviation; SD = 47.19), while the range was 100 (100/1988; 5.03%) to 245 (245/1988; 12.32%), with the highest reported in 2009. By species, whales, dolphins, dugongs, and sea turtles had the most strandings in 2010, 2013, 2012, and 2014, respectively (Figure 1d).

### 3.2. Spatial and Temporal Analyses

Most strandings occurred along the coast of the Gulf of Thailand (1132/1988; 56.94%), while the remaining strandings occurred along the coast of the Andaman Sea (856/1988; 43.06%). Significant clustering (*p* < 0.01) of stranded marine endangered species was detected in the following 14 provinces: Satun, Trang, Phuket, Phang Nga, Phatthalung, Nakhon Si Thammarat, Surat Thani, Chumphon, Prachuap Khiri Khun, Samut Sakhon, Samut Prakan, Chonburi, Rayong, and Trat (Figure 1b). Kernel analysis also showed a high density of stranded marine endangered species in similar areas. In this regard, the biggest clustering and highest density of stranded animals was observed in Phuket (Figure 1c). In addition, the appearance of significant clusters (*p* < 0.01) was similar in the rainy, summer, and winter seasons from 2006 to 2015 (Figure 2). However, in 2006, 2007, 2009, and 2012 to 2014, more stranding clusters appeared in the Gulf of Thailand in the winter season than in the other seasons.

The total number of stranded marine endangered species was 884 (884/1988; 44.47%), 495 (495/1988; 24.90%), and 609 (609/1988; 30.63%) for the rainy, summer, and winter seasons, respectively. Consequently, the average numbers of stranded animal locations over the 10 years by season were 88.40 (SD = 24.30), 49.50 (SD = 14.91), and 60.90 (SD = 17.93) for the rainy, summer, and winter seasons, respectively. One-way ANOVA tests showed that the number of stranding locations differed significantly across seasons (*p* < 0.05, F = 10.58). A multiple comparisons test showed that the average number of animals stranded in the rainy season was significantly higher than that in the summer and winter (*p* < 0.05) (Figure 3a). We found that the total numbers of animals stranded in the coastal area of the Andaman Sea were 516 (516/1988; 25.96%), 163 (163/1988; 8.20%), and 177 (177/1988; 8.90%) for the rainy, summer, and winter seasons, respectively. For animals stranded in the coastal area of the Gulf of Thailand, the total numbers were 368 (368/1988; 18.51%), 332 (332/1988; 16.70%), and 432 (432/1988; 21.73%) for the rainy, summer, and winter seasons, respectively. In this regard, Pearson’s Chi-Square results showed a significant association (*p* < 0.05, χ^2^ = 153.90) between the number of stranded animals across coastal locations and seasons (Figure 3b).

### 3.3. Factors Associated with Marine Stranding

At the provincial level, the length of coastline (*Rs* = 0.5419, *p*-value = 0.067) and the number of islands (*Rs* = 0.5902, *p*-value = 0.268) were not significantly associated with the number of stranded animals.

## 4. Discussion

Of the 1988 stranding events, whales, dolphins, dugongs, and sea turtles appeared in both the Andaman Sea and the Gulf of Thailand. The average number of stranded marine endangered species was 198.80 animals/year (SD = 47.19), with a range of 100–245, which was the highest in 2009. Of the 24 provinces, significant clusters (*p* < 0.01) of stranded marine endangered species were detected in 14 provinces, with Phuket having the largest cluster and highest stranding density. In addition, the average number of stranded animals in the rainy season was significantly higher than that in summer and winter (*p* < 0.05).

In this study, sea turtles showed the highest frequency of stranding. This is because sea turtles tend to stay in coral reefs or along coastlines as their preferred feeding and developmental habitat, and this can often lead to stranding. In addition, the high abundance of sea turtles leads to more turtle strandings than cetacean strandings. Moreover, human activities, such as high fishing activity, possibly increase the likelihood of stranding of marine animals. A study in the Atlantic Ocean concluded that sea turtle stranding is influenced by individual, health, and oceanographic factors [18]. Furthermore, sea turtle bycatch also influences the stranding rate [19,20]. Previous studies in Thailand reported that anthropogenic factors, such as plastic waste or fishery activities, were important factors that led to sea turtle stranding [12,21]. In this regard, further research is needed to determine the factor with the greatest effect on sea turtle stranding in Thailand. We found that green turtles were the most commonly stranded species of sea turtles, and this possibly mirrors their behaviour of foraging in shallow waters near the coast or islands. However, research should also consider the effect of bycatch on strandings. Dugongs had the lowest frequency of stranding because of their low numbers in the sea of Thailand, putting them at risk of extinction. Therefore, rescuing stranded dugongs is important. For stranded whales and dolphins, 13 and 11 types by common names, respectively, were found. This indicates that the sea of Thailand has an abundance of whales and dolphins that makes it possible to determine species diversity. Previous reports have shown that fishing gear led to the mortality of cetaceans. This is an important factor associated with the loss of endangered species [14]. Consequently, the establishment or improvement of surveillance and monitoring systems is an important approach to rescuing stranded animals. Surveillance and monitoring locations can be selected based on the clustering observed in this study. However, the effectiveness of a surveillance system requires the cooperation of all stakeholders from different sectors, including local residents.

The annual average number of stranded endangered marine species in Thailand from 2006 to 2015 was 198.80 animals/year. This is a high number of strandings compared to that in other countries, such as the Philippines. In the Philippines, the annual average number of marine mammal strandings is 59 animals per year [22]. The Philippines has a longer coastline, which is likely to be a high-risk area for stranding. However, a good surveillance system can help reduce the number of stranded animals. A previous study indicated that coastal topography, climate, animal density, and the effectiveness of the country’s stranding surveillance system are factors affecting marine stranding events [23]. Therefore, the creation or improvement of the surveillance system in Thailand is one of the best approaches to monitoring and reducing the problem of stranded marine endangered species by preventing or intercepting strandings and giving timely attention to stranded individuals. Furthermore, with an effective surveillance system, the relevant authorities can monitor and implement measures to help live-stranding animals immediately.

In this study, we could not determine the exact cause of stranding. However, some records show living and dead stranded animals with external wounds caused by fishing tools or vessels, such as propellers, nets, ropes, or hooks. Although these tools and vessels may affect the stranding of the animals, finding the primary cause of a stranding must come from a complete necropsy on a fresh carcass. This highlights the necessity of establishing an effective surveillance system for stranded animals. In addition, from our discussions with DMCR officers, we learned that plastic debris and/or garbage foam have been found inside the carcasses. This suggests that human-caused waste affects the health of marine animals and is one of the causes of marine stranding in Thailand. Regarding plastic waste, previous studies have reported that plastics pose a risk to marine animals and humans in coastal areas [24] and plastic-induced novel ecosystems of the coastal ocean by seagrasses [25]. This may negatively affect the livelihood of marine animals.

Furthermore, the results of the cluster analysis and kernel density estimation showed consistency in the location of the largest stranding cluster and highest stranding density. Coastal areas in Phuket showed the largest cluster and highest stranding density (Figure 1b,c). Therefore, this location (including other clustered and high-density locations) should be a hotspot for stranding surveillance and monitoring. In this way, it is possible to rescue stranded animals immediately. A previous study also reported the hotspot location for marine mammal stranding. Researchers have suggested that this type of analysis will improve the stranding response, particularly during periods of large strandings and with limited response resources [26]. In addition, our cluster analysis showed that some clustered areas covered several provinces. Thus, in some cases, measures to reduce animal stranding require the cooperation of many provinces. However, stranding clusters might differ based on spatial and temporal occurrence. Our results showed more clustering in some winters in the Gulf of Thailand than in other seasons, especially in 2013 (Figure 2). In Thailand, monsoons start to have a very high influence in November in the Gulf of Thailand, with possible annual differences in severity [27,28]. This may be an important factor contributing to the higher stranding rates in some years during the study period. However, further research is needed to determine the degree of association between monsoons and the stranding of endangered marine species in the Gulf of Thailand.

The coastline connecting to the Gulf of Thailand showed more strandings than the coastline connecting to the Andaman Sea. This is because the coastline connecting to the Andaman Sea is shorter than that connecting to the Gulf of Thailand. However, the largest clustering appeared along the coast of the Andaman Sea (at the area of Phuket). The coastline of the Andaman Sea has more geographic factors that are conducive to marine stranding. It has a steep terrain with indented cliffs, while the appearance of both small and large bays, including many islands, is also an important factor in stranding. Moreover, the characteristics of coral reefs in this area contribute to strandings [29,30]. In addition, during the rainy season, the Andaman Sea had a higher number of strandings than the Gulf of Thailand (Figure 3b), which influences the total number of stranded animals in this season (Figure 3a). During the rainy season, southwest monsoons in the Andaman Sea appear from May to October [31], and this period is characterised by high surface waves, high sea levels, and turbulence in shallow waters [32,33,34,35]. These factors affect animal strandings, and further research is warranted to determine the degree of influence. However, these data on stranding are limited to Thailand. If we had data on stranded marine endangered species in neighbouring countries, the severity of stranding in the Andaman area might have been higher than that in the Gulf of Thailand. Therefore, neighbouring countries in this region should jointly study the stranding of endangered marine species.

In each province, the coastline length and the number of islands showed no significant association with the number of strandings. However, we did not include factors associated with anthropogenic activities or other environmental factors in our analysis. For anthropogenic factors, we already obtained data on the number of registered fishing vessels. However, the data do not contain fishing locations, routing, or GPS recording systems. Therefore, it was difficult to analyse the data without bias. In this regard, the Department of Fisheries of Thailand stipulates that licensed fishing vessels with a vessel size of 30 gross tons must be equipped with a vessel monitoring system [36]. Thus, for further analysis, these types of data may be analysed in association with stranding data. However, small fishing vessels are not forced to install these systems, which creates a gap in stranding data due to fishing activities by small vessels. In this regard, the effect of fishing activities by small vessels on stranding requires further investigation. Regarding environmental factors, we used data on pH, temperature, salinity, and dissolved oxygen, measured at the surface of the sea of Thailand. However, as the data only included a few measurement locations and did not cover the entire coastal and ocean area, it became difficult to assess the relationship between these factors and the number of strandings without bias. Therefore, the implementation of automatic recording systems with remote sensing technology at several locations in the coastal area and the sea of Thailand is needed for effective surveillance and monitoring of the aforementioned factors (including other important factors, such as heavy metal residues).

Notably, in our cluster analysis, we selected a grid size of 5 × 5 mapping units, which we considered appropriate for topography and coastlines. The selected grid size was fitted with areas of Thailand on both sides of the coast. We attempted to analyse grids larger and smaller than the selected grid size and found that some grids covered both sides of the coast or the appearance of clusters had low sensitivity. Therefore, the selected grid size was suitable for our study.

## 5. Conclusions

This study showed that the annual number of stranded marine endangered species in Thailand was much higher than that in other countries, such as the Philippines. More strandings occurred in the rainy season than in other seasons, with the Gulf of Thailand having more strandings than the coast of the Andaman Sea. We suggest that surveillance and monitoring systems should be established to focus on stranded animals through spatial and temporal analyses. Several locations that are stranding hot spots require the implementation of measures or plans to reduce the number of strandings. Moreover, the causes of stranding and related factors should be studied based on surveillance and monitoring systems.

## Figures and Tables

**Figure 1 biology-12-00448-f001:**
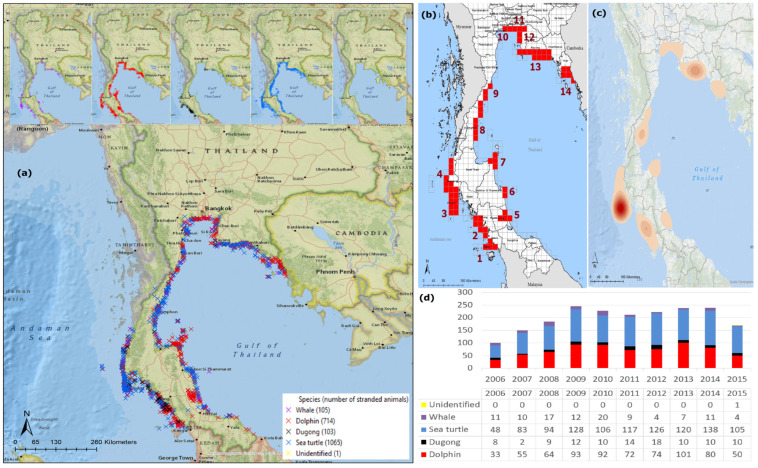
(**a**) Map of Thailand showing the locations of endangered marine species stranded in Thailand from 2006 to 2015; (**b**) Hot spots (red squares) of animal stranding according to cluster analysis. Significant clusters (*p* < 0.01) were detected in the following 14 provinces: (1) Satun, (2) Trang, (3) Phuket, (4) Phang Nga, (5) Phatthalung, (6) Nakhon Si Thammarat, (7) Surat Thani, (8) Chumphon, (9) Prachuap Khiri Khun, (10) Samut Sakhon, (11) Samut Prakan, (12) Chonburi, (13) Rayong, and (14) Trat; (**c**) Density of stranded animals from high (red colour) to low (orange colour) according to Kernel density analysis; (**d**) Number of stranded animals per year by species.

**Figure 2 biology-12-00448-f002:**
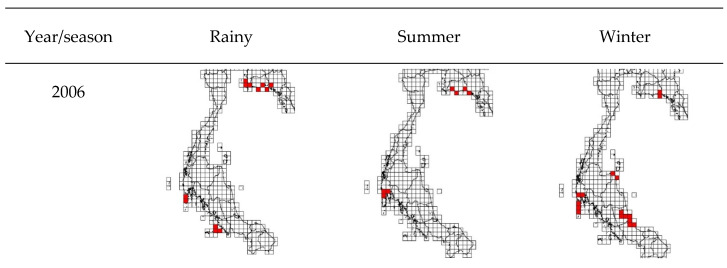
Spatial and temporal clustering of endangered marine species stranded from 2006 to 2015. The red square symbol indicates significant hotspot locations using the Anselin Local Moran’s I (*p* < 0.01).

**Figure 3 biology-12-00448-f003:**
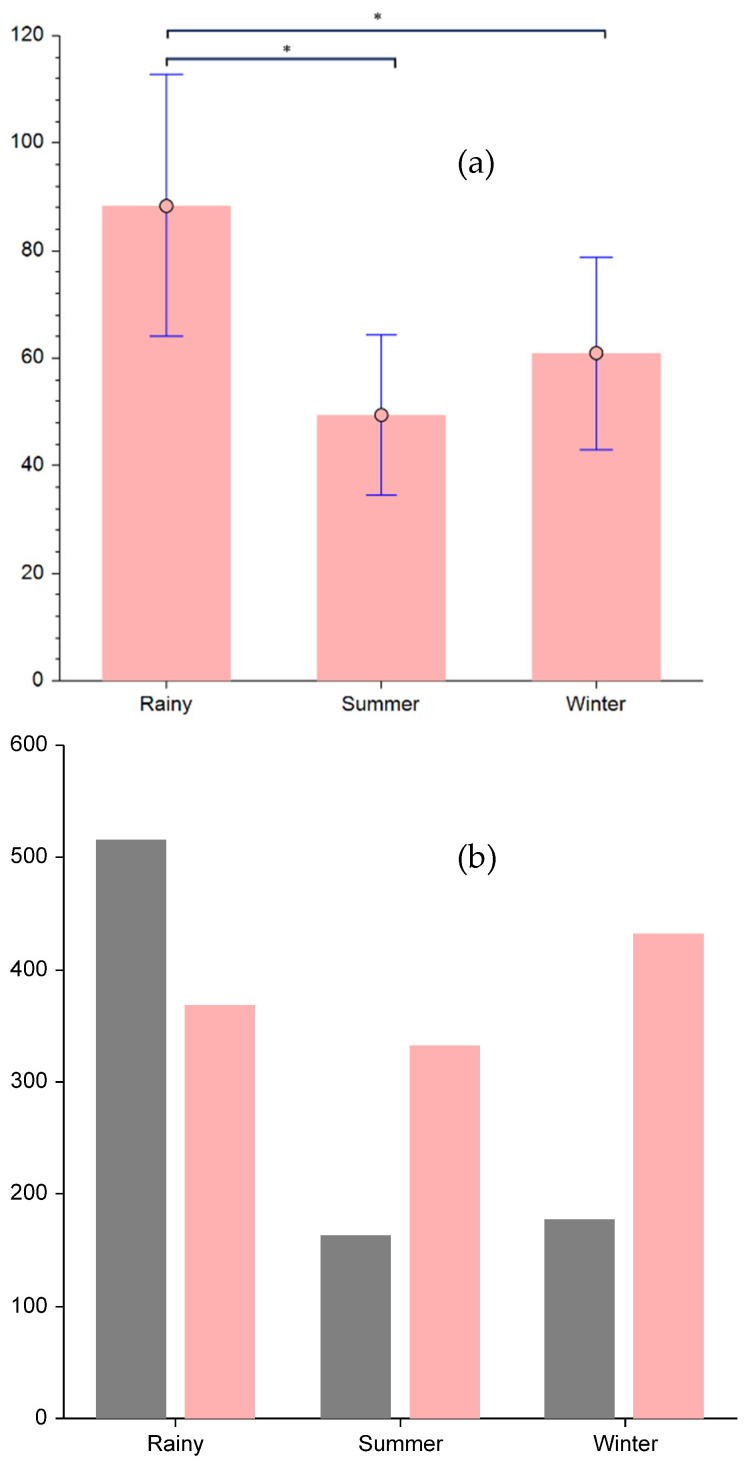
Bar graphs with error bars representing the number of endangered marine species across three seasons. (**a**) Comparison of mean and standard deviation of the number of stranded animals by season. Results of one-way analysis of variance (ANOVA) and Bonferroni test show a comparison of the number of stranded marine endangered species (*Y*-axis) by season (*X*-axis). Asterisks indicate that the rainy season was significantly different from summer and winter (*p* < 0.05). (**b**) The bar graph presents the number of stranded animals (*Y*-axis) by season and coastal location (*X*-axis). The grey bars show the number of stranded animals in the Andaman Sea, while the pink bars show the number of stranded animals in the Gulf of Thailand. In addition, Pearson’s chi-squared test showed a significant association between coastal location and season (*p* < 0.05).

## Data Availability

The datasets generated and/or analysed during the current study are available from the corresponding author upon reasonable request.

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
