# Peer review of "Spatial, Temporal, and Geographical Factors Associated with Stranded Marine Endangered Species in Thailand during 2006–2015"

_biology, 2023, doi:10.3390/biology12030448_

Round 1

Reviewer 1 Report

Congratulations on this interesting work which is very useful in terms of marine vertebrates' health surveillance and stranding response in the study area as well as an example for other regions.

Please, find below a few suggestions to be improved.

Line 129-131 The reason why sea turtles are the most recorded stranded species is also due to the abundance of the species compared to cetaceans. I would suggest adding a consideration about it in these lines.

Line 162-166 The cause of death could be determined only with a complete necropsy on a fresh carcass (condition code of the carcass from 1-3). Otherwise, it should be referred only to as "human interaction evidence/signs/findings". I would suggest reformulating these lines in order to better present the discussion.

Line 239: The authors report that the number of the annual strandings is "very high". I would suggest reporting a reference to compare this number as high.

Author Response

Response to Reviewer 1

Congratulations on this interesting work which is very useful in terms of marine vertebrates' health surveillance and stranding response in the study area as well as an example for other regions.

Response: We would like to thank the reviewer for the insightful review of our manuscript. Please see our responses to the comments below. In addition, all changes in the revised manuscript are marked using tracked changes.

Moreover, this revised manuscript has been edited for language by a native English speaker. This way, we believe that the revised manuscript now meets the journal’s publication requirements.

Please, find below a few suggestions to be improved.

Line 129-131 The reason why sea turtles are the most recorded stranded species is also due to the abundance of the species compared to cetaceans. I would suggest adding a consideration about it in these lines.

Response: We agree with the comment. We have accordingly revised the manuscript (lines 143–145, pg. 11).

Line 162-166 The cause of death could be determined only with a complete necropsy on a fresh carcass (condition code of the carcass from 1-3). Otherwise, it should be referred only to as "human interaction evidence/signs/findings". I would suggest reformulating these lines in order to better present the discussion.

Response: We agreed with the reviewer’s comments and have rewritten this part of the manuscript accordingly (lines 184–186, pg. 11).

Line 239: The authors report that the number of the annual strandings is "very high". I would suggest reporting a reference to compare this number as high.

Response: We have added supporting evidence to our discussion with references (lines 161–166, pg. 10). Thus, we did not add references to the conclusion section; instead, we have improved the writing of this section (line 267–268, pg. 13).

*******************

Reviewer 2 Report

This is an interesting paper that highlights the need for a surveillance system to record strandings of marine vertebrates (mammals and reptiles) on the coasts of Thailand, have a faster response to attend the live strandings, and determine the cause of death of carcasses.

The authors did a great job introducing the problem and explaining their methods, but these could be improved to provide more detailed information on the possible causes of the strandings. The results and the discussion also need additional work. Some of their statements lack scientific support, and you could fix this by giving another thorough review of similar work done in other regions of the world. I strongly suggest including a second analysis to compare the number of strandings with the fisheries' seasons, especially those fisheries that use nets or long-lines.

I also have specific comments that will hopefully help improve your manuscript:

Lines 61-63: the scientific names of sea turtles should be in italics too.

Line 97: In the citation, you misspelled the word institute.

On page 5, the line numbers reset to 1. The following comments are from pages 5 to 16.

Lines 15-16: Did the authors consider analyzing the data based on fishing seasons? There might be a correlation with the number of strandings.

Lines: 64-65: These data are interesting. It seems that strandings are increasing through the years. This information is worth exploring for this paper as it also highlights the importance of characterizing the pressures associated with the strandings.

Lines 93-94: But what does this mean? Could you explain this association between the coastal location and season more?

Lines 130-131: I strongly suggest changing this statement. Sea turtles do not move to make feeding easier. They move because that is where their feeding and developmental habitats exist.

Lines 131-132: But strandings are influenced by bycatch too, and globally. Several papers deal with this problem: https://scholar.google.com/scholar?hl=en&as_sdt=0%2C5&q=sea+turtle+fisheries+bycatch&btnG=

Lines 135-136: This can be tested easily if you compare the number of strandings against the fisheries season. Do fishermen rely on nets or long-lines to catch their prey?

Line 137: Not necessarily. The stranding could be related to bycatch, as is often the case in other regions.

Lines 140-141: This statement is incorrect. Low genetic diversity does not drive low numbers of individuals; it is the opposite. Low numbers of reproductive individuals result in low genetic diversity.

Lines 150-151: Did you notice changes in the number of strandings through the years? These data could also help understand the dimension of the problem.

Lines 155-157: Are you talking about the number of RECORDED stranding events? Please, specify.

Lines 158-159: This sentence is confusing. Are these systems supposed to intercept the animals before they reach the beach? Or do you mean that with a system in place, the number of dead stranded animals will decrease because they will get timely attention?

Line 194: What about oceanographic factors? Currents can play a substantial role in where the strandings will occur, especially in animals that are sick and cannot move.

Line 215: Agreed, but there must be a fishing season too. In some countries, different fishing seasons depend on the target species: lobster, tuna, octopus. If the exact location of the vessels is unknown, there should be a general area where fishing occurs. It is worth exploring if you have that information.

Lines 260 and on: The references are not in the same text format as the rest of the manuscript. Also, many of the citations are technical reports. This is acceptable, but you need more scientific references to support your discussions. There is a vast number of papers dealing with strandings, their causes, and stranding networks around the world that might help strengthen your discussion and conclusions.

In figure 1a, adding an inset with a macro location of the study area would be better for the readers. In this figure, using more contrasting colors for the markers would be better. Whales and dugongs are hard to see.

In figure 3, Letters a) and b) are missing in the graphs. In the text, the color of the bars should say grey, not black.

Author Response

Response to Reviewer 2

This is an interesting paper that highlights the need for a surveillance system to record strandings of marine vertebrates (mammals and reptiles) on the coasts of Thailand, have a faster response to attend the live strandings, and determine the cause of death of carcasses.

Response: We would like to thank the reviewer for the insightful review of our manuscript. Please see our detailed responses to the reviewer’s comments below. In addition, all changes in the revised manuscript are marked using tracked changes.

Moreover, this revised manuscript has been edited for language by a native English speaker. This way, we believe that the revised manuscript now meets the journal’s publication requirements.

The authors did a great job introducing the problem and explaining their methods, but these could be improved to provide more detailed information on the possible causes of the strandings. The results and the discussion also need additional work. Some of their statements lack scientific support, and you could fix this by giving another thorough review of similar work done in other regions of the world. I strongly suggest including a second analysis to compare the number of strandings with the fisheries' seasons, especially those fisheries that use nets or long-lines.

Response: We have added detailed information on the possible causes of strandings in the Introduction using references to related studies (lines 93–96, pg. 2–3). In addition, we have added references to marine strandings in the Discussion (lines 160–162 and 179–181, pg. 11). Regarding the relationship between fisheries seasons and the number of strandings: in Thailand, the fisheries season is year-round, with fishermen always using similar equipment for different marine species. Therefore, it is difficult to classify the fisheries season.

I also have specific comments that will hopefully help improve your manuscript:

Lines 61-63: the scientific names of sea turtles should be in italics too.

Response: We have corrected all relevant parts according to this reviewer’s comment (lines 74–85, pg. 2).

Line 97: In the citation, you misspelled the word institute.

Response: Please note that we have changed our citation style according to the journal’s requirements.

On page 5, the line numbers reset to 1. The following comments are from pages 5 to 16.

Response: We apologise for this mistake, which came about when the publisher developed a new layout of the manuscript. However, we have carefully checked and corrected our manuscript by following the reviewer’s comments. We are sure that all of our responses are in relation to the reviewer’s comments.

Lines 15-16: Did the authors consider analyzing the data based on fishing seasons? There might be a correlation with the number of strandings.

Response: As the fisheries season occurs year-round in Thailand, it is difficult to classify a specific fisheries season.

Lines: 64-65: These data are interesting. It seems that strandings are increasing through the years. This information is worth exploring for this paper as it also highlights the importance of characterizing the pressures associated with the strandings.

Response: According to Figure 1d, the number of strandings has increased slightly, but with some variation. To come to a clearer conclusion, we suggest a follow-up study is needed for further investigation.

Lines 93-94: But what does this mean? Could you explain this association between the coastal location and season more?

Response: We have improved our manuscript according to this comment (lines 98–100, pg. 7).

Lines 130-131: I strongly suggest changing this statement. Sea turtles do not move to make feeding easier. They move because that is where their feeding and developmental habitats exist.

Response: We agree and have accordingly revised our manuscript (lines 141–145, pg. 11).

Lines 131-132: But strandings are influenced by bycatch too, and globally. Several papers deal with this problem: https://scholar.google.com/scholar?hl=en&as_sdt=0%2C5&q=sea+turtle+fisheries+bycatch&btnG=

Response: We agree with this comment. We have improved our manuscript according to this comment (lines 146–147, pg. 11).

Lines 135-136: This can be tested easily if you compare the number of strandings against the fisheries season. Do fishermen rely on nets or long-lines to catch their prey?

Response: It is difficult to classify the fisheries season in Thailand, as we have no related data and fishing occurs year-round. Fishermen always fish for marine animals with the same set of equipment but may catch different marine species.

Line 137: Not necessarily. The stranding could be related to bycatch, as is often the case in other regions.

Response: We agree and have updated our manuscript accordingly (lines 153–154, pg. 11).

Lines 140-141: This statement is incorrect. Low genetic diversity does not drive low numbers of individuals; it is the opposite. Low numbers of reproductive individuals result in low genetic diversity.

Response: We apologise for our mistake. We have checked the cited article and have decided to remove this sentence from our manuscript.

Lines 150-151: Did you notice changes in the number of strandings through the years? These data could also help understand the dimension of the problem.

Response: We attempted to compare the annual average number of strandings with those of neighbouring countries to investigate the status of marine strandings in Thailand (lines 168–169, pg. 11). We also considered the variation in the number of stranded animals and found that there may be an increasing trend. However, the number of stranded animals in the last year of the analysed period (2015) showed a decrease from the previous year (Figure 1d). It could be that there are other factors that we have not yet studied. Therefore, it is possible that we may need a longer-period study to see trends more clearly.

Lines 155-157: Are you talking about the number of RECORDED stranding events? Please, specify.

Response: We apologise for this confusion. We have improved our manuscript to clarify that the sentence refers to factors associated with marine stranding (line 175–176, pg. 11).

Lines 158-159: This sentence is confusing. Are these systems supposed to intercept the animals before they reach the beach? Or do you mean that with a system in place, the number of dead stranded animals will decrease because they will get timely attention?

Response: We apologise for the confusion. In this sentence we are referring to both issues. With an effective surveillance system in place, we can monitor very closely and help live-stranded animals immediately. We have accordingly revised the manuscript (lines 179–181, pg. 11).

Line 194: What about oceanographic factors? Currents can play a substantial role in where the strandings will occur, especially in animals that are sick and cannot move.

Response: We have mentioned geographical factors that affect marine strandings in lines 217–224, pg. 12.

Line 215: Agreed, but there must be a fishing season too. In some countries, different fishing seasons depend on the target species: lobster, tuna, octopus. If the exact location of the vessels is unknown, there should be a general area where fishing occurs. It is worth exploring if you have that information.

Response: In Thailand, the fisheries season occurs year-round for the majority of marine species, including fish, shrimp, and squid. Thus, it is difficult to classify a specific fisheries season.

Lines 260 and on: The references are not in the same text format as the rest of the manuscript. Also, many of the citations are technical reports. This is acceptable, but you need more scientific references to support your discussions. There is a vast number of papers dealing with strandings, their causes, and stranding networks around the world that might help strengthen your discussion and conclusions.

Response: We have improved our manuscript style and referencing format. Four new references were added according to the reviewer’s comments.

In figure 1a, adding an inset with a macro location of the study area would be better for the readers. In this figure, using more contrasting colors for the markers would be better. Whales and dugongs are hard to see.

Response: We have improved this figure according to the reviewer’s comments (Figure 1, pg. 5).

In figure 3, Letters a) and b) are missing in the graphs. In the text, the color of the bars should say grey, not black.

Response: We apologise for these mistakes and have corrected them in the revised manuscript (Figure 3 and legend, pg. 10).

**************

Round 2

Reviewer 2 Report

Dear authors, thank you for your responses to my comments and for the changes in your manuscript. After reviewing the second version of your manuscript, I have additional comments:

1) Based on your response: We have added detailed information on the possible causes of strandings in the Introduction using references to related studies (lines 93–96, pg. 2–3). In addition, we have added references to marine strandings in the Discussion (lines 160–162 and 179–181, pg. 11). Regarding the relationship between fisheries seasons and the number of stranding events: in Thailand, the fisheries season is year-round, with fishermen always using similar equipment for different marine species. Therefore, it is difficult to classify the fisheries season.

The introduction and methods section should include the information highlighted in yellow as it is important background for your study. This justifies why the analysis only includes seasonality and not fisheries seasons. 

Lines 141-143. This statement still needs some work. They way it is written gives the idea that sea turtles strand because they are usually found in coral reefs or near the coastlines. While this is partially true, authors should mentioned that these habitats are also associated with high fishing activity, which increases the probability of incidental capture and stranding.

Lines 351 - 354. These citations have the same title, please check.

Author Response

Response to Reviewer 2 (round 2)

Dear authors, thank you for your responses to my comments and for the changes in your manuscript. After reviewing the second version of your manuscript, I have additional comments:

Response: We would like to thank the reviewer for the insightful review of our manuscript. Please see our detailed responses to the reviewer’s comments below. In addition, all changes in the revised manuscript are marked using tracked changes.

1) Based on your response: We have added detailed information on the possible causes of strandings in the Introduction using references to related studies (lines 93–96, pg. 2–3). In addition, we have added references to marine strandings in the Discussion (lines 160–162 and 179–181, pg. 11). Regarding the relationship between fisheries seasons and the number of stranding events: in Thailand, the fisheries season is year-round, with fishermen always using similar equipment for different marine species. Therefore, it is difficult to classify the fisheries season.

The introduction and methods section should include the information highlighted in yellow as it is important background for your study. This justifies why the analysis only includes seasonality and not fisheries seasons. 

Response: We agree and have accordingly revised our manuscript (lines 123–126, pg. 3).

Lines 141-143. This statement still needs some work. They way it is written gives the idea that sea turtles strand because they are usually found in coral reefs or near the coastlines. While this is partially true, authors should mentioned that these habitats are also associated with high fishing activity, which increases the probability of incidental capture and stranding.

Response: We agree and have accordingly revised our manuscript (lines 145–146, pg. 11).

Lines 351 - 354. These citations have the same title, please check.

Response: Thank you very much for pointing this out. We have reviewed the citations and found that Olson et al., 2018 is an abstract from a poster presentation in a conference, whereas the other one is a journal article. Nevertheless, we have decided to remove the reference citing the abstract presentation to avoid redundancy. Please see lines 352–353, pg. 15.

Round 3

Reviewer 2 Report

Dear authors,

Thank you for your patience and for considering my recommendations. I am satisfied with the changes made to the document, and I do not have additional comments.